# The composite governance mechanisms and sustainable economic performance of Pakistan's textile industry

Asima Saleem[1]*, Maqsood Ahmad Sandhu[2], Aijaz Mustafa Hashmi[3], Aisha Jamil[4]

1 National University of Modern Languages, Islamabad, Pakistan, 2 College of Business and Economics, United Arab Emirates University, Al Ain, UAE, 3 Faculty of Management Sciences, National University of Modern Languages, Islamabad, Pakistan, 4 University of Lahore, Lahore, Pakistan

* asimas009@gmail.com

**Data Availability Statement:** Data supporting to this article is uploaded in Supporting information.

**Funding:** The author(s) received no specific funding for this work.

## Abstract

This study examines the influence of corporate governance mechanisms on the sustainable economic performance of the textile industry in Pakistan. Performance is measured by using a global reporting initiative (GRI 201–1), an economic performance approach. The composite corporate governance mechanism index (CGMI) construction consists of five sub-indices: board of directors, disclosure and transparency, audit committee, shareholder rights, and remuneration committee. The empirical analysis demonstrated the consistency of the fixed effect model through the Hausman model specification test for data from textile firms listed on the Pakistan Stock Exchange from 2008 to 2023. However, the investigation has invoked the Generalized Method of Moments (GMM) estimation model to check its robustness. The findings indicated the positive empirical relationship between the composite CGMI and sustainable economic performance, supported by modified theoretical logics: agency theory, stewardship theory, stakeholder theory, resource-based theory, and transaction-cost theory. The findings support policymakers, regulators, and managers in executing appropriate governance mechanisms with reduced agency cost and transaction costs and improve sustainable performance.

## 1. Introduction

In the ever-evolving landscape of contemporary markets, the pursuit of success has become more intense, necessitating not only prosperity for companies but also ensure lasting viability [1]. In this context, sustainability has emerged as a pivotal factor for successful organizations, as underscored by more than 90% of CEOs who recognize its significance [2]. The concept of sustainability, which originated from Brundtland's seminal work in 1987, revolves around meeting present-day needs without compromising the ability of future generations to meet their own needs [3]. This principle has gained further impetus with the United Nations Development Program (UNDP), which makes corporate sustainability pivotal to attaining global sustainability goals by 2030 [4]. Specifically, UNDP's Sustainability Goal 12.6 urges corporations, particularly large and transnational entities, to embrace sustainable practices and

**Competing interests:** The authors have declared that no competing interests exist.

integrate comprehensive sustainability disclosures into their annual reports [5]. As emphasized by the UN's Sustainable Development Goals, the corporate sector's pivotal role in upholding sustainability is undeniable.

Suharyono and Hs [6], stated that there are numerous international standards for measuring performance impacts, such as the United Nations Global Compact (UNGC), the Environmental Program (UNEP), and the Global Reporting Initiative (GRI). Among these performance measurement guidelines, GRI Standards are the most widely applied procedure for corporate sustainability. Sustainable performance has become a paramount concern in developing nations. However, research in this domain remains scarce [7]. To address this gap, the Global Reporting Initiative (GRI) has provided globally recognized sustainability standards, serving as a yardstick for gauging performance impact, enhancing reporting quality, and guiding optimal corporate decisions. While these standards have been extensively adopted, their contextual nuances, particularly in emerging economies, remain underexplored [8].

Governance principles, as formulated by the Organization for Economic Cooperation and Development (OECD) in 2004, encapsulate regulatory frameworks, policies, and mechanisms crafted by authorities to safeguard the interests of stakeholders and capital providers. These principles should align with an organization's overarching objectives [9]. Although the relationship between governance practices and organizational performance has drawn substantial attention over the past two decades, such as [10–13]. The evaluation of corporate governance in the context of sustainable performance still needs exploration. The existing research on sustainable performance impacts has mainly focused on advanced economies, leaving a dearth of insights in the context of developing nations such as India and Pakistan [14].

Previous studies considered a single metric framework for measuring sustainable performance, focusing on the perspective of GRI Sustainability standards, namely Economic Social and Governance (ESG), referenced studies include [15–18]. The in-depth exploration of various dimensions of GRI Sustainability improves the credibility of sustainable practices. They highlight the fact that the preferable approach embraces economic sustainability in an era when every firm is concerned to improve its sustainable economic performance [19]. The use of a composite index decreases the outliers and the aggregation process covers wider dimensions. Only a few studies have applied a composite index for both corporate governance and sustainable performance, e.g. Suharyono and Hs, Achim et al. [6, 20]. Some studies have applied agency theory to evaluate the corporate governance-performance relationship. Bui and Krajcsak [21], argued that numerous critical corporate governance theories have evolved over the years, and the empirical findings need to be elaborated on given the presence of multidimensional theoretical underpinnings.

In the Pakistani context, CG awareness dates back only a few decades. The evolution of the CG Code in Pakistan, initiated by the Security and Exchange Commission of Pakistan (SECP) in 2002 and subsequently revised, led to changes in compliance dynamics. Early on, adherence to these codes was optional, but evidence suggests that companies that followed these codes outperformed those that did not [22]. Gradually, these codes became mandatory, with subsequent revisions emphasizing such aspects as board structure, transparency, and regulatory compliance. Despite the growing interest in CG and its impact, research in Pakistan has yielded conflicting results, necessitating further investigation [22]. Miao et al. [23], argued that Pakistan has a comparatively simple legal regulatory system with inadequate control mechanisms, and requires both a unified Corporate Governance code and sound CG practices before it can foster certain corporate approaches and obtain optimal results. They maintained that Pakistan seems to treat CG based on personal relationships as the norm and that this may eventually result in financial ruin. Younas et al. [24], however, specifically focusing on the

corporate sector in Pakistan, showed that the adoption of the right governance practices positively influences corporate financial distress by reducing risk.

The textile sector is a leading manufacturing industry in Pakistan and is of particular interest. It holds a prominent position as an exporter of textile products in Asia, contributing substantially to GDP and employment [25]. As evidenced by recent data from the Pakistan Economic Survey (PES), the textile industry has lately experienced robust growth and export performance, underscoring its strategic importance [26]. This industry is the 8th biggest textile product exporter in Asia. The PES 2021–2022, surveying large-scale manufacturing industries (LSM), reported 18.16 percent growth in the textile sector (www.finance.gov.pk). The textile industry has shown an upswing among local and international markets, with a 34 percent growth in garment production during the year 2022. According to the Pakistan Bureau of Statistics (PBS), the textile sector saw a 25.4 percent increase in exports by 2022 [26]. The textile industry has a lengthy production chain, with a contribution of approximately one-fourth of the industrial value addition. This sector employs about 40 percent of the industrial workforce. Aside from cyclical and seasonal fluctuations, this sector has maintained a total share of nearly 61.24 percent of domestic exports.

The textile industry in Pakistan comprises three subsectors listed on the Stock Exchange: textile composites, textile spinning, and textile weaving. The sector has received the maximum share of total financing of 202.9 billion, 94.6 percent of which has been financed by the industry itself. Tahir and Anuar [25], found that the contribution of the textile sector is approximately 46 percent of the aggregate manufacturing and 8.5 percent contribution of the annual GDP of Pakistan. Considering the latest data from PBS, Pakistan's earnings from apparel and textile exports increased from \$2.933 billion in 2021 to \$3.056 billion in 2022. According to PES 2021–2022, this industry is experiencing steady growth due to underlying oil price fluctuations, limited technology and investments, and the energy crisis; however, increased financing, imports of textile machines, and high exports show substantial forwarding in the textile industry (www.finance.gov.pk). The top managers of various businesses—Sarah Textiles, Taj Company, ENGRO Group, and Crescent Bank—after their failures have realized the importance of governance mechanisms for not only their management but also for avoiding future losses to the companies [22]. Very few studies evaluated CGI in the context of the textile sector in Pakistan, such as Tahir et al. [27]. Given these research gaps and contextual dynamics, the following core research question emerges:

Does a composite governance mechanism index influence the sustainable economic performance of Pakistan's textile industry? The primary objective of the present study is to investigate the influence of the composite governance mechanism index on the sustainable economic performance of the textile industry in Pakistan.

In light of the gaps described above, the notable contributions made by this paper to the existing literature are as follows: 1) in line with the referenced studies [20, 21, 28]; 2) the study modifies the theoretical understanding of the CG-sustainable performance relationship. The study also modifies the findings of the CG-performance literature on sectoral contexts within developing economies as a result of examining the application of CGMI to sustainable economic performance in Pakistan's textile sector [21, 28]; 3) the study applies GRI 201–1, an economic performance approach for evaluating sustainable performance impacts; 4) The study seeks to provide a robust theoretical foundation for the connection between CGMI and sustainable economic performance through the implications of five modified theoretical logics [21].

The remainder of the study is structured as follows: Section 2 presents the literature review highlighting theoretical contributions, concepts, and empirical relationships; Section 3 discusses a comprehensive research methodology; Section 4 shows a thorough data analysis and an interpretation of the findings; Section 5 presents a discussion and some conclusions.

## 2. Literature review

### 2.1. Theoretical lens

The traditionalist concept of agency theory argues that a conflict of alignment exists in the principal-agent relationship, which negatively affects shareholders' wealth maximization, and produces a residual loss when the firm's revenues are transferred into managerial discretion [29]. Lindkvist and Saric, Jensen and Meckling [30, 31], calculated that the estimated agency costs for publicly traded corporations resulted in a 16% decline from the benchmark, or $1432 million, in firm value. As a result of the residual loss caused by agency conflicts and the rise in agency costs, the value of companies significantly decreased [31]. Bui and Krajcsak [21], highlighted agency theory among the theoretical underpinnings of corporate governance but indicated that the right governance mechanisms ensure that firm managers behave in their shareholders' best interests. Referenced studies including Benson and Ganda [32] and Munir et al. [16], applied agency theory to establish the empirical relation between CGM, financial performance, and sustainability performance.

The present study applied agency theory to minimize agency conflict, and agency cost by applying the right governance mechanisms consisting of the board of directors, audit committee, disclosure and transparency, remuneration committee, and shareholder's rights, and improve the sustainable performance of the firms.

Bui & Krajcsak [21], described managers as stewards of the organization, and asserted that the presence of trust, collaboration, and cooperation between shareholders and managers decreases agency costs and enhances the firm's performance. Stewardship theory is unique in creating increased trust between managers and shareholders [22, 33]. According to Lambright [34], shareholders' interests as stewards are aligned with the wish to maximize the long-term stewardship of the company. These writers applied agency and stewardship theories to evaluate the CG-performance relationship [22]. The study applied stewardship theory to consider trust, collaboration, and cooperation between shareholders and managers, on the one hand, and enhanced sustainable performance, on the other.

Stakeholder theory argues that organizations are accountable to the interests of both stakeholders and shareholders and effective governance mechanisms should align with their best interests [21]. Munir et al. [16], applied stakeholder theory and described the link between CG and sustainable performance in these terms: "the framework of governance mechanisms shall warrant the security of stakeholder's rights by incorporating the social, environmental, and economic issues into the organization strategies and practices", by Galbreath [35]. Benson and Ganda [32], applied stakeholder theory to evaluate the dynamic interactions between CGM, financial performance, and corporate sustainability. A combination of existing traditional theories was applied to align the interests of the corporate stakeholders and improve sustainable performance. Saleem et al. [36], applied stakeholder theory for establishing the link between board diversity and sustainability.

The resource-based conception (RBT) highlights the view that firm performance depends on the resources commanded by the firm [37]. RBT supported the view that a company's operations and success are based on internal and external resources, namely labor, raw materials, and capital [21]. Accessibility to these resources depends upon the company's link with internal and external stakeholders and is significantly monitored in the course of effective corporate governance. James and Joseph [38], argued that internal stakeholders facilitate the access to resources that positively influence firm performance. Bui and Krajcsak [21], applied the RBV and demonstrated a significant empirical relationship between corporate governance performance impacts. Gardazi et al. [39], argued that directors manage resources through expertise and skills, improved decision-making procedures, and ultimately corporate performance. The

study applied governance mechanisms for the effective use of internal and external resources and to enhance sustainable performance.

According to the transaction cost theory, corporate governance frameworks depend on the overall effect of internal and external business transactions [40]. The theory holds that corporate governance ensures the effective and efficient fulfillment of transactions by stakeholders. Bui and Krajcsak [21], argued that firms engage in many transactions, such as employee hiring and contracts with suppliers, that incur negotiating and monitoring costs. They show empirically that the establishment of appropriate governance mechanisms can prune these costs through the establishment of clear procedures and rules for transactions. The study applied transaction cost theory to minimize the transaction cost by clarifying the transaction rules between managers and shareholders.

The studies also applied the Upper Echelons Theory (UET) that considered the impact of top-level management characteristics on firm outcomes [41]. Phan and Duong [42], applied UET and proved a significant positive influence of management characteristics on corporate performance, namely board size, gender diversity, and CEO's knowledge capability.

## 2.2. Theoretical concepts

**2.2.1. Corporate governance mechanisms.**   Existing studies such as Mensah and Adams [43] have systematically developed CG, illustrating it as a process to maximize value for stakeholders that affects the direction and governance of enterprise affairs. It specifically focuses on enhancing a firm's prosperity and accountability and the achievement over the long run of organizational objectives. Comprehensive governance practices are concerned, for example, with board size, audit committees, remuneration, shareholding and ownership, and governance disclosures [44].

Tahir et al. [45], documented board composition as a measure of board size and board independence. Arora and Bodhanwala [46], discussed board size as the number of directors serving on a corporate board and highlighted it as the top decision-making forum directing the efforts of senior management. The dynamic composition of the Board of Directors (BOD) ensures the presence of sound CG practices [47]. Aguilera et al. [48], highlighted the BOD as an important internal governance mechanism. Studies such as Fariha et al. [47] and Guney et al. [49], proxied board structure in terms of board size, diversity, CEO duality, and independence. Javaid and Saboor [50], created a sub-index of board structure along the dimensions of board size, the ratio of outside directors, dual CEOs, executive directors, number of board meetings and effectiveness, and the presence of a chief financial officer.

The audit committee (AC) is an extended representation of the organization's board, defined by Rahman et al. [51]. The AC audits financial statements at regular intervals and aims to improve the credibility of financial reports [50]. The AC is responsible for designing, overseeing, and implementing the financial reporting practices of the company and is thus able to ensure better governance procedures [51]. Salloum et al. [52], argued that an effective AC is based on two attributes: the quality and independence of the AC director. The writers whom they cite, such as Fariha et al. [47] and Rahman et al [51], applied a range of attributes to measure AC characteristics: AC size, meetings, independence, diligence, and external audit quality. Farooq et al. [22], created a CGI with 29 governance elements, divided into ownership structure, compensation, board committees, and audit committees.

Information disclosure and transparency (D&T) are important attributes for improving the quality of governance mechanisms [53]. Bushman et al. [54], conceptualized information D&T as the openness of corporate-related information to stakeholders and outside investors. Temiz [55], claimed that D&T is a crucial element of CG, specifically focusing on the settlement of

information gaps. D&T significantly contributes to enhancing corporate operational efficiency, documented by Lai et al. [56]. According to Ashbaugh et al. [57], effective information D&T practices reduce agency costs by limiting management's chances to behave opportunistically and managing their actions. Truong et al. [58], created a D&T index consisting of three sub-indices: financial D&T, business legal ownership, and investor relations, and the composition of the board and management. Javaid and Saboor [50], created a CGI with a sub-index of disclosure, such as disclosure of CG elements, remuneration, audit committee, shareholding categories, executive members' ownership, and financial reports.

According to Mintah [59] the remuneration committee (RC) is a considerable sub-group of the board, with the specific duties of scrutinizing compensation-related board decisions: salary, bonus, rewards, superannuation payments, share options, commission, health insurance, pension contributions, and shareholder profit sharing. Harymawan et al. [60], argued that effective RC reduces agency expenses and information asymmetry. Kanapathippillai et al. [61], demonstrated that the presence of quality RC minimizes corporate risk levels through the escalation of voluntary disclosures. RC diligence and independence are key attributes that improve the RC quality. Harymawan et al. [60], measured RC using two common proxies: executive remuneration and the board of directors' remuneration.

The investors' ownership concentration (OC) is a considerable driver of good governance mechanisms, ultimately resulting in performance improvement and efficiency gains [62]. According to agency theory, OC can help in the effective monitoring of corporate operations management, the alleviation of agency costs, and the solving of information problems resulting in reasonable performance improvement. The most common measurement for OC is the shareholders' percentage of shareholdings [63]. Guluma [64], measured OC as the total percentage of the top 10 block holdings. OC is an effective governance practice that occupies a position in the CGI Arora & Bodhanwala [46]. Javaid and Saboor [50] created an ownership structure sub-index consisting of seven elements: block holders, institutional ownership, direct ownership, managerial ownership, voting share percentage, ownership concentration, and family ownership.

## 2.3. Empirical relationships

**2.3.1. CGM and sustainability performance.** The extensive literature on CG practices and financial effectiveness is based on two performance metrics: one accounting-based and the other market-based, studies include [21, 28, 65]. They applied agency theory to show that right governance mechanisms help to align the interests of the stakeholders and ultimately enhance financial performance. The cited studies evaluate the interdependencies between CG practices and sustainability performance, specifically focusing on reporting and the quality of the disclosure perspectives, e.g., Tjahjadi et al. [66], evaluated the impact of right governance practices on the social, environmental, and economic sustainability performance of listed non-finance firms of the Indonesian Stock market for 2013–2017. The study applied six versions of the GRI reporting standards and contributed to UET and agency theory by demonstrating the significant influence of the size of the board of commissioners, their educational background, and that of the top management team on sustainable performance through multiple regression. Shrivastava and Addas [15] made clear the significant influence of CG tools on Environmental, Social, and Governance (ESG) disclosure, the proportion of the corporate board that attended meetings, and the proportion of the board that was independent: as a measure of sustainability performance, the governance disclosure score measured for CG and ESG disclosure were applied. Studies have also determined the individual influence of diverse CGM on sustainability, but the results obtained are inconclusive, conflicting, debatable, and irreconcilable

[32]. They mentioned a theoretical gap owing to scant research in the context of sustainability studies, specifically for developing economies.

Munir et al. [16], argued that the controversial findings on the interdependencies between CG, business performance, and sustainability performance could together develop a robust conceptual framework for evaluating these relationships. By contributing to agency theory, RBV, and stakeholder theory the study revealed the favorable effects of the board of governance attributes on the sustainability of the energy sector (Gardazi et al. [39]. Naciti [67], scrutinized the impact of board composition on the sustainable performance of 362 companies from 46 countries over the period 2013–2016 by applying the Generalized Method of Moments (GMM) estimation. Aligning with stakeholder theory and agency theory, her findings indicated that companies supporting high board diversity and CEO duality are highly sustainable, and having more independent directors negatively affects sustainability, as measured by the sustainability score of ESG. She argued that only a handful of studies have evaluated the interrelation between board structure and the company's sustainability procedures. Munir et al. [16], employed Structural Equation Modeling (SEM) to show the favorable effects of CG on the sustainability of listed companies in the Australian security market (ASX). The sustainability performance index takes into account social, economic, environmental, health, and safety management systems. These writers examined the impact of CG on enterprise performance by considering non-financial organizations listed on the PSX from 2003 to 2014 Hussain et al. [81]. The study supported agency theory by demonstrating the favorable effects of CG practices on business performance through GMM Arellano-Bond dynamic panel-data estimation.

Few studies have considered the impact of CGI on sustainability performance. Sar [68], assessed the effect of CGM on the ESG sustainability performance of listed Indian FMCG companies obtained from the CMIE-PROWESS database. The questionnaire survey showed the favorable effects of CGM on ESG sustainability. Saleem et al. [36], investigated the contribution of board diversity towards the achievement of sustainability of agile non-profitable organizations. The study applied SEM analysis and results supported the relationship between stakeholder's theory of corporate governance and sustainability. These writers examined the effect of CG procedures on the corporate sustainability of listed South African companies from 2010 to 2020 by applying ESG, ROA, and ROE as sustainability performance measures [32]. Their study applied Dynamic Ordinary Least Square (OLS) and Fully Modified OLS, and achieved significant but inconclusive results about sustainability. They argued that dynamic and healthy interactions between CG, corporate performance, and corporate sustainability would result in value creation. Achim et al. [20], evaluated the effect of quality governance practices on the long-term development of 185 nations from 2005 to 2020, through panel regression analysis. The findings indicated that high-quality governance practices contribute positively to sustainable development. Madaleno and Vieira [68], examined the effect of gender CG practices on the business performance and sustainability practices of listed companies in Spain and Portugal from to 2010–2017. The study took the GMM approach and established the favorable effects of gender governance on business finances and sustainability. Irshad et al. [69], contributed to agency theory, RBV, and stakeholder theory by establishing the significant positive relationship between CG and the sustainable environmental performance of listed USA firms for 2004–2017. Kwarteng et al. [70], aligning the study with agency theory, proved the significant positive effect of all the board characteristics on the sustainable performance of a list of Sub-Saharan African firms from 2010 to 2019. Board characteristics included board independence, board tenure, board size, board sub-committees, board gender, board-CEO duality, board educational competencies, CEO power, board meetings frequency, and foreign BOD. In line with the above theoretical findings, the following hypothesis may be proposed:

$H_1$: The composite governance mechanism index has a positive influence on the sustainable economic performance of Pakistan's textile industry.

**2.3.2. Firm size, firm age, and GRI sustainability.** To determine the indirect influence on GRI Sustainability, a recent study used firm size and age as control variables. Considering the modeling difficulties, the evidence studies Guluma [64] and Arora and Bodhanwala [46], employed firm age and firm size to evaluate the influence of CGM on corporate performance. According to Guluma [64], large firms have major agency problems and need strong governance mechanisms, which ultimately influence firm performance. Guluma [64], documented several measures for firm size: revenue volume, market capitalization, number of employees, firm age, and how long an enterprise has been operating in the market since larger firms tend to have a longer operating history and this significantly contributes to better performance. Referenced studies such as [28, 46, 71], revealed a significant controlled effect of firm size and firm age on CGM-corporate financial performance.

## 3. Methodology

The methodology of the present study incorporated four important descriptions: detailed sample selection and data sources, measurement variables, modeling equation for the proposed models, and description of estimation techniques.

### 3.1. Sample selection and data sources

To determine the impact of CGM on GRI sustainability, we took for our initial study sample 124 textile firms listed on Pakistan's Security Market (PSX), These textile firms were divided into three sub-sectors: textile composite (53 companies), textile spinning (61 companies), and textile weaving (10 companies). For the final sample selection, a dual sample selection criterion was applied: 1) the elimination of firms that became delinquent and questioned due to the lack of financial information, as disclosed by the annual reports of listed textile enterprises; and, 2) the elimination of firms that were never listed during the entire study period, as highlighted by Elston [72] and Saleem & Hashmi [73]. Finally, 26 companies were eliminated, and the panel consisted of 98 firms listed on PSX and considered for the evaluation of their sustainable performance: 43 companies that were textile composites, 49 companies that specialized in textile spinning, and 6 companies that specialized in textile weaving.

The study period covered 15 years between 2008 and 2023. The range of data shows a high crisis period following the 2008 financial crisis, another following the 2020 pandemic of COVID-19, financial constraints and financial frictions between 2015 and 2021, and other macroeconomic turbulences, in line with Abdulzahra et al. [74].

The datasets for both GRI sustainability measures and CGI measures were collected from the annual reports of publicly traded textile companies. The study considered Pakistan because it is one of the GRI member states [75], and the choice of the textile sector as the study sample is based on the following characteristics: 1) the textile sector consists of a higher proportion of firms than the other firms listed on the Pakistan security market (psx.com.pk); and 2) this sector provides a comprehensive understanding of the entire textile sector in Pakistan as composed of three sub-sectors, namely, textile composite, textile spinning, and textile weaving firms; 3) the textile sector is of significant importance to the economy [76].

## 3.2. Descriptions of the variables

To measure sustainable economic performance, this study implemented three operationalized components of GRI 201–1, the sustainability of the economic performance approach: direct economic value generated (DEG), economic value distributed (DED), and economic value retained (DER), adopted from GRI [77]. DEG is measured through revenues obtained in the form of net income, net revenues from asset sales, collections of interests, and dividends. DED is measured through salaries and other employee benefits, such as insurance, pensions, interest-free loans, operational expenses, taxes, payments of interest on loans, dividends, and community support programs. DER is the difference between the economic value generated and the economic value distributed. The study proceeded stepwise: 1) calculation of the aggregate score for DEG and DED dimensions that presents the overall use of the dimensions, and 2) calculation of the DER by taking the difference between DEG and DED, which represents the GRI sustainability performance score, similar to Saleem & Hashmi [73]. The score for GRI sustainability performance was calculated as

$$GRI\ Sustainability\ Performance = DER = DEG - DED$$

According to Black et al. [78], CG indices are more useful for determining rising firm value, specifically in developing markets. The comprehensive CGMI, adopted from Munisi and Randoy [79], had five sub-indices, namely, the board of directors (BOD), audit committee (AC), Disclosure and Transparency (D&T), Remuneration Committee (RC), and shareholders' rights (SR). This study applied an equal-weighted index creation method, which considers the allocation of equal (100%) weights to each item of the sub-index, as discussed in Javaid & Saboor [50]. The validity of each statement used to measure the sub-indices was verified from the sample textile companies' annual reports. A 'Yes' response showed the presence of an element and was coded 1 and a 'No' shows the absence of an element and was coded 0. The study proceeded stepwise to determine the index score: 1) The aggregate score for each sub-index on a firm-specific basis was calculated by averaging the value of all items in each sub-index, and 2) the aggregate score for CGMI was then determined by taking the average of the score of all five sub-indices, bearing in mind the following formula:

$$CGMI = \frac{Aggregate\ score\ of\ sub - index\ for\ each\ firm}{Total\ number\ of\ items}$$

Table 1 presents the description of 39 items that were incorporated for measuring sub-indices, classified as BOD (7 elements), AC (5 elements), D&T (20 elements), RC (4 elements), and SR (3 elements), adopted from Munisi & Randoy [79]. However, Akbar [80], highlighted the fact that Pakistan lags in the adoption of CG principles and strategies.

The study applied two control variables: firm size and firm age. Many current studies measure firm size by logarithm (ln) of total assets, and firm age by years of operating experience, referenced by Arora and Bodhanwala [46], Do et al [71], Affes & Jarboui [28].

## 3.3. Empirical model

The empirical model consists of CGMI as an explanatory variable, GRI as a response variable for measuring sustainable performance effects, and firm age (F_age) and firm size (Fs) as control variables. In the equation, 't' is the number of years, $i$' the indexes firm, '$e_{it}$' the random

**Table 1. Measurement variables of CGMI.**

| Sub Index | Measurement items |
|---|---|
| 1. Board of Directors | 1. Separate people serve as the CEO and board chair.<br>2. The enterprise chairman is a non-executive director.<br>3. The company identifies the classes of directors.<br>4. The non-executive directors made up at least two thirds of the board.<br>5. The business displays the frequency of board-organized meetings.<br>6. The board has a committee for CG.<br>7. The board has a nomination committee. |
| 2. Audit Committee | 8. The firm has an AC.<br>9. The committee chairman is a non-executive director.<br>10. Non-executive directors make up the entire committee.<br>11. The board chairperson is not a member or a chairman of the audit committee<br>12. The company displays the frequency of committee-organized meetings. |
| 3. Disclosure and Transparency | 13. IFRS (International Financial Reporting Standards) are adopted.<br>14. The firm discloses the RC composition.<br>15. The firm discloses the AC composition.<br>16. The firm discloses the total gross compensation of all directors.<br>17. The firm discloses the CEO's compensation.<br>18. The firm discloses the professional/employment backgrounds of its top officials.<br>19. The firm discloses the educational backgrounds of its superior officers.<br>20. The firm discloses the top management team's compensation.<br>21. The firm discloses the professional/employment backgrounds of its directors.<br>22. The firm discloses the educational backgrounds of its directors.<br>23. The firm discloses the director's ages.<br>24. The firm discloses each director's appointment date.<br>25. The firm considers independent auditors as the big four audit firms.<br>26. The firm discloses its annual report within three months of the year's conclusion.<br>27. The firm discloses its stock prices and security market performance.<br>28. The firm reveals its share of OC.<br>29. The firm discloses its affirmation of effective governance practices.<br>30. The firm discloses the critical evaluation of financial results.<br>31. The firm discloses an analysis of five-year financial patterns.<br>32. The firm discloses the CSR activities reports. |
| 4. Remuneration Committee | 33. The business has an RC.<br>34. The chair of the committee is a non-executive director.<br>35. The entire committee consists of non-executive directors.<br>36. The firm discloses the regularity of committee-organized meetings. |
| 5. Shareholders' Rights | 37. The firm adheres to the equal rights principle i.e., one share-one vote.<br>38. The firm appoints all directors yearly.<br>39. The business demonstrates the implication of Proxy voting. |

Source: (Munisi and Randoy, 2013).

error term and '$\alpha$' the intercept.

$$GRI_{it} = \alpha_0 + \alpha_1\, CGMI_{it} + \alpha_2\, FSIZE_{it} + \alpha_3\, FAGE_{it} + e_{it}$$

## 3.4. Panel econometrics

The study applied commonly used panel regression approaches, Random Effect (RE) and Fixed Effect (FE) statistical methods, following the assumption that unobserved heterogeneity would not correlate with the explanatory variables. The implication of panel ordinary least squares (OLS) statistics gave biased and inconsistent results that may have been due to endogeneity issues, consistent with Arora and Bodhanwala [46]. FE and RE estimation models were

applied to test the relationship of individual-specific effects with the explanatory variables by controlling all the time-invariant differences between the entities and reducing biases in the estimated coefficients of the model because of omitted time-invariant characteristics. The Hausman test was then applied to determine whether the RE model provided a consistent and unbiased estimate.

The study also applied GMM through Arellano-Bond dynamic panel-data estimation to check robustness, a method adopted by Hussain et al. [81]. The GMM model is an effective strategy for addressing panel data with an endogeneity bias [82]. For the regression analysis, the applied preliminary statistics included unit root tests and the Durbin-Watson autocorrelation test.

## 4. Results and discussion

### 4.1. Descriptive statistics

Table 2 shows the summary statistics of the comprehensive metrics of CGMI and GRI Sustainability for the 1,092 observations. The mean values for BOD, AC, and D&T were 0.66, 0.62, and 0.64, respectively, indicating the implementation of CG practices in companies. The mean value for RC was low. i.e., 0.34, indicating less-established remuneration committees in industrial sectors. SR has a mean value of 0.53, indicating less consideration for shareholder rights in Pakistan than elsewhere. The mean value of CGMI is also less than 0.56, indicating less importance being given to governance mechanisms in Pakistani industries. The mean value of GRI is 8.93, ranging between a minimum of 7.02 and a maximum of 11.92, showing high DEG by firms through dividends, interest collections, and operational efficiency. The standard deviation of all the variables is less than 1, indicating that the data points are closer to the average value.

The Fs has a mean value of 8.22, ranging between 5.73 and 10.30; the maximum value reveals that a larger firm possessed assets totaling Rs 35 million, while the minimum value indicates that one of the sample enterprises with a lower size owned assets totaling only Rs 537,821. F-age has a mean value of 14.51 ranging between 14.34 and 17.3, showing that the sample firms have a minimum of 14 years of operational history.

### 4.2. Preliminary statistics

**4.2.1. Serial correlation.** The study applied Durbin-Watson (D-B) serial correlation statistics to test the null hypotheses $H_0$: no autocorrelation and $H_1$: autocorrelation exists. Wooldridge [83], stated that the value for D-W statistics ranges between 0 and 4, with 0 as a positive correlation, 4 as a negative correlation, and 2 as no correlation. Azami et al. [84], reported that

**Table 2. Basic statistics of Pakistan.**

| Variable | Observations | Mean | Minimum | Maximum | Standard deviation |
|---|---|---|---|---|---|
| GRI | 1,092 | 8.929 | 7.022 | 11.923 | 0.356 |
| BOD | 1,092 | 0.662 | 0.234 | 0.681 | 0.243 |
| AC | 1,092 | 0.628 | 0.144 | 0.543 | 0.261 |
| D&T | 1,092 | 0.648 | 0.164 | 0.512 | 0.451 |
| RC | 1,092 | 0.343 | 0.132 | 0.453 | 0.572 |
| SR | 1,092 | 0.533 | 0.092 | 0.582 | 0.418 |
| CGMI | 1,092 | 0.563 | 0.151 | 0.692 | 0.542 |
| Fs | 1,092 | 8.220 | 5.734 | 10.301 | 0.525 |
| F_age | 1,092 | 14.515 | 14.348 | 17.364 | 0.604 |

**Table 3. Results of the D-W serial correlation test.**

| Variable | Observations | D-W d-statistic |
|----------|--------------|-----------------|
| Residuals | 1,092 | (6, 14) 2.223 |

the acceptance value for D-W lies between 1.5 and 2.5. In Table 3, the Durbin-Watson statistic value, which measures the level of autocorrelation in the residuals, shows an acceptance value of 2.22 within the reported range of 6 and 14. Hence, the null hypothesis of no autocorrelation is not rejected.

**4.2.2. Panel unit root test.** The study applied three-panel unit root tests for stationarity analysis among the variables: the Levin-Lin-Chu (LLC), Im-Pesaran-Shin (IPS), and Fischer Augmented Dickey-Fuller (ADF) tests, to test $H_1$: Panels are stationary and $H_0$: Panels contain unit roots. A value below 0.05 significance level indicated rejection of $H_0$, hence stationarity is present. Table 4 presents the results of the LLC, IPS, and ADF tests for GRI, BOD, AC, D&T, RC, SR, and CGMI, which consist of t-statistics above ±1.69 and p-values below 0.05, indicating that the acceptance of $H_1$, i-e panels is stationary.

**4.2.3. VIF multicollinearity test.** Variance Inflation Factor (VIF) statistics were applied to estimate multicollinearity. The standard significance range for VIF was less than 5, stated by Wooldridge [83]. The study found that the mean VIF for predictor variables was 1.78, hence the multicollinearity issue was satisfied.

## 4.3. Panel regression analysis

**4.3.1. Fixed effect, random effect estimation.** This study applied FE and RE statistics to determine the causality of CGMI on GRI Sustainability Performance. The RE statistics tested the $H_0$, that there was 'no interrelationship between the explanatory variables and the error term,' and FE statistics evaluated the $H_1$, that 'an interrelationship existed between the explanatory variables and the error term,' by Yngman et al. [85]. The Hausman test estimates the null hypothesis $H_0$, that the difference between the fixed effects and the random effects estimators is not systematic. The p-values below 0.05 rejecting $H_0$ and indicating that the FE is more valid and has values greater than 0.05 fail to reject $H_0$, and indicate the validity of the RE model.

Table 5 shows the results of the FE estimation model for Pakistan 's textile industry. Table 6 shows the results of the RE estimation model, and Table 8 shows the results of the GMM estimation model. FE, RE, and GMM statistics obtained a significant p-value of 0.000, i.e., less than 0.05, and the t-statistics above ±1.69, indicate acceptance of Hypothesis $H_1$: that CGMI positively influences the GRI Sustainability Performance of the textile sector in Pakistan. GMM shows the strong significant effect of L.GRI on GRI. Both Fs and Fage demonstrated a

**Table 4. Results of LLC, IPS, and ADF test.**

| Variable | LLC | | IPS | | ADF | |
|----------|------|------|------|------|------|------|
| | t-statistics | p-value | t-statistics | p-value | t-statistics | p-value |
| GRI | -12.610 | 0.000 | -12.608 | 0.000 | 32.938 | 0.000 |
| BOD | -6.271 | 0.000 | -5.796 | 0.000 | 9.812 | 0.000 |
| AC | -14.526 | 0.000 | -11.081 | 0.000 | 27.389 | 0.000 |
| D&T | -16.893 | 0.000 | -11.866 | 0.000 | 31.911 | 0.000 |
| RC | -11.270 | 0.000 | -10.113 | 0.000 | 22.865 | 0.000 |
| SR | -5.126 | 0.000 | -7.771 | 0.000 | 20.525 | 0.000 |
| CGMI | -11.394 | 0.000 | -9.014 | 0.000 | 10.774 | 0.000 |

**Table 5. Results of FE model estimates.**

| Variable | FE Estimation | | | | | |
|---|---|---|---|---|---|---|
| | Coefficient | Standard error | t-value | p-value | [95% Conf. Interval] | |
| CGMI | 0.156 | 0.047 | 3.32 | 0.001 | 0.064 | 0.248 |
| Fs | 0.197 | 0.055 | 3.61 | 0.000 | 0.089 | 0.305 |
| F_age | 0.141 | 0.052 | 2.71 | 0.007 | 0.039 | 0.243 |
| Constant | 0.150 | 0.045 | 3.29 | 0.001 | 0.060 | 0.239 |
| R-square | 0.354 | | | | | |
| F-statistics | 17.03 | | | | | |
| Prob > F | 0.000 | | | | | |
| Hausman test | 0.000 | | | | | |

**Table 6. Results of RE model estimates.**

| Variable | RE Estimation | | | | | |
|---|---|---|---|---|---|---|
| | Coefficient | Standard error | t-value | p-value | [95% Conf. Interval] | |
| CGMI | 0.235 | 0.040 | 5.95 | 0.000 | 0.158 | 0.313 |
| Fs | 0.254 | 0.047 | 5.37 | 0.000 | 0.161 | 0.347 |
| F_age | 0.202 | 0.047 | 4.31 | 0.000 | 0.110 | 0.294 |
| Constant | 0.131 | 0.043 | 3.05 | 0.002 | 0.047 | 0.216 |
| R-square | 0.359 | | | | | |
| Wald chi2 | 192.6 | | | | | |
| Prob > F | 0.000 | | | | | |
| Hausman test | 0.000 | | | | | |

statistically significant favorable controlled effect on GRI Sustainability, with p-values <0.05. The confidence intervals do not include a zero value; standard error shows values closer to 0 and a positive beta coefficient indicates a significant positive relationship. The p-value for the Hausman model specification test is 0.05, supporting the validity of the fixed effect. F-statistics are 17.3 showing acceptable model fitness and the R-squared value presents an explained variance of 35%. Wald chi2 at 196.3 shows acceptable model fitness with CGMI significantly contributing to explain the variation in the GRI Sustainability.

**4.3.2. Endogeneity test.** Endogeneity exists due to the correlation between a predictor variable and the error term, possibly due to measurement error, omitted variables, sample selection bias or simultaneity [86]. The issue can be resolved by incorporating instrument variables or lagged variables [87]. The study tested the $H_0$: the presence of a significant association between the error term and a predictor variable, at p-values above 0.05 and t-statistics less than ±1.69 significance. Table 7 shows the results of endogeneity tests, i.e., the FE regression estimates between the error term and a predictor variable, reported p-value >0.05, and t-statistics below ±1.69, satisfying the endogeneity issue.

**Table 7. Results of endogeneity test.**

| Variable | Coefficient | Standard error | t-value | p-value | [95% Conf. Interval] | |
|---|---|---|---|---|---|---|
| Error term | 0.034 | 0.024 | 1.41 | 0.159 | -0.013 | 0.080 |
| Fs | 0.346 | 0.082 | 2.97 | 0.003 | 0.016 | 0.033 |
| F_age | 0.235 | 0.071 | 2.23 | 0.025 | 0.013 | 0.080 |
| Constant | 0.668 | 0.075 | 8.96 | 0.040 | 0.522 | 0.815 |

**Table 8. Results of GMM model estimates.**

| Variable | GMM Estimation | | | | | |
|---|---|---|---|---|---|---|
| | Coefficient | Standard error | t-value | p-value | [95% Conf. Interval] | |
| L.GRI | 0.538 | 0.197 | 2.73 | 0.006 | 0.152 | 0.924 |
| CGMI | 0.317 | 0.051 | 6.19 | 0.000 | -0.417 | -0.217 |
| Fs | 0.262 | 0.064 | 4.07 | 0.000 | 0.136 | 0.388 |
| F_age | 0.394 | 0.062 | 3.57 | 0.000 | -0.127 | -0.215 |
| Constant | 0.422 | 0.071 | 1.91 | 0.040 | 0.213 | 0.493 |
| Wald chi2 | 65.01 | | | | | |
| Prob > F | 0.000 | | | | | |
| Arellano-Bond test AR (2) | 0.352 | | | | | |
| Hensen's J test | 0.084 | | | | | |
| Sargan test | 0.066 | | | | | |

**4.3.3. GMM estimation.** Hussain et al. [81], argued that the GMM is the best solution to the dilemma of endogeneity and unobservable heterogeneity, specifically focusing on the relationship between CG and business performance. The study applied GMM through the Arellano-Bond dynamic panel-data model to estimate the causal effect of CGMI on GRI Sustainability. Lagged GRI (L.GRI) is created as an instrument variable. Iftikhar et al. [88], documented GMM estimation as the best method to resolve the endogeneity issue through the panel GMM estimation model.

Table 8 shows the results of the GMM Estimation model. GMM statistics for lagged GRI and CGMI have a significant p-value of 0.000. i.e., less than 0.05, and t-statistics above ±1.69, indicating acceptance of Hypothesis $H_1$: CGMI positively influences the GRI Sustainability Performance of the textile sector in Pakistan. Both Fs and Fage demonstrated a statistically significant favorable controlled effect on GRI Sustainability, with p-values<0.05. The confidence intervals do not include a zero value, standard error shows values closer to 0 and a positive beta coefficient indicates a significant positive relationship. The p-value for the Hausman model specification test is 0.05, supporting the validity of the fixed effect. Wald chi2 shows acceptable model fitness with CGMI significantly contributing to explain the variation in the GRI Sustainability and the R-squared value presenting an explained variance of 36%. The Arellano-Bond test at second order difference with a p-value above 0.05 significance level shows the need to reject $H_1$: serial autocorrelation is present. Hensen's J test and Sargan's test have p-values above 0.05, indicating the validity of the instruments.

These results are in line with the studies [16, 32, 68], who found a positive influence of CGM on sustainable performance. Javeed and Azeem [89], argued that an advanced CG system is crucial for a company's long-term existence. The findings mainly contributed to agency theory by reducing the agency cost and agency conflicts between managers and stakeholders [21, 28, 65]. Significant sustainable performance is supported by the management of resources by skilled BOD, aligned with Gardazi et al. [39]. The right governance mechanism enhances trust, while collaboration and cooperation between shareholders and managers decrease agency costs and enhance sustainable performance [22, 21].

## 5. Discussion and conclusion

This study examined the effect of CGMI on the GRI Sustainability performance of Pakistan's textile sector. The study implied a comprehensive CGM and created an index based on five sub-indices by applying the equal-weighted index creation method. Corporate sustainability

was measured by incorporating GRI 201–1, an economic performance application consisting of DEG, DED, and DER. The study applied FE and RE panel regression models to the data gathered from the yearly statements of textile companies listed on the PSX from 2007 to 2022. The GMM model was applied to check the robustness. The statistics indicated the significant positive influence of CGMI on the sustainability of the GRI in this industry. The finding concluded that the right governance mechanisms consisting of BOD, AC, financial reporting D&T, RC, and SR, significantly improve the Firm's sustainable performance. The combination of modified theoretical logic from agency theory, stewardship theory, stakeholder theory, resource-based theory, and transaction-cost theory significantly supported the empirical relationship between CGMI and GRI Sustainability.

The positive CGM-GRI sustainable performance relationship aligns with existing studies: Madaleno and Vieira [82], Benson and Ganda [32], Bui & Krajcsak [21]. The impact of positive sustainable performance indicates the contribution of the right governance mechanisms in reducing the agency cost, transaction cost, minimizing the conflict of interests between stakeholders, and building trust, collaboration, and cooperation between shareholders and managers, and effective use of internal and external resources. The findings mainly contributed to agency theory [16, 32]. Advanced theories are also supported, namely, stewardship theory [22], stakeholders' theory [32, 35], and RBV [21]. The transaction cost perspective is least explored regarding cost minimization with clear transaction rules [21].

The findings from the study serve to modify the literature with indications that the textile sector in Pakistan is lagging in having the right governance mechanisms, in line with Farooq et al. [22]. The scores obtained from the indices indicate the need for improvement in governance quality for this industry. RC and SR obtained very low scores on summary statistics, but effective regulatory measures could support their right implementation making a guaranteed contribution to the goal of maximizing shareholder value and better firm performance.

The conclusions drawn from the current study have considerable policy implications, especially focusing on good governance regulations for the corporate sector. Farooq et al. [22], highlighted that findings from studies of this type will be supportive of researchers, academicians, practitioners, and regulators who faithfully demonstrate the true connection between governance practices and corporate performance. The study upholds the comprehensive realization of the contribution of governance mechanisms for sustaining the economic performance of the textile industry and catalyzes the role of policymakers and regulators in establishing and implementing policies that support the sustainable performance of good governance practices. The current study provides guidance for managers of organizations regarding better decision-making in consideration of stakeholders' and shareholders' interests, which are mainly governed by regulations and laws concerning governance mechanisms.

With all the foregoing discussion, the present study still has some limitations, which could be the scope of future studies. First, this study applied equal-weighted index creation for corporate governance mechanisms, whereas future studies could take a value-weighted index creation approach. Second, the study findings specifically contribute to the textile sector only; future scope can be extended to other important sectors, such as oil and gas, pharmaceuticals, and the automotive industry. In addition to contextual limitations, the current study is limited to Pakistan, and there is scope in the future for exploring the surrounding integrated territories: the Asian economies and the SAARC nations. Fourth, the study applied the GRI sustainability economic performance approach, while future studies can expand the scope of social and environmental sustainability standards to determine performance effects. Finally, the study implemented traditional random effect and fixed effect panel regression statistics but future studies could consider the generalized least squares (GLS) Models: GMM, ARDL, and FGLS models.

## Supporting information

**S1 Data.**
(XLSX)

## Acknowledgments

The authors are most grateful to the anonymous reviewers for their constructive and helpful comments, which helped considerably to improve the presentation of the paper.

## Author Contributions

**Conceptualization:** Aijaz Mustafa Hashmi.

**Data curation:** Asima Saleem.

**Formal analysis:** Asima Saleem.

**Methodology:** Asima Saleem.

**Validation:** Maqsood Ahmad Sandhu.

**Writing – review & editing:** Maqsood Ahmad Sandhu, Aisha Jamil.

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
