## [Decision Letter · Decision Letter 0]

28 Jun 2024

PONE-D-24-17105Composite Governance Mechanisms and Sustainable Economic Performance of Pakistan Textile IndustryPLOS ONE

Dear Dr. Saleem,

Thank you for submitting your manuscript to PLOS ONE. After careful consideration, we feel that it has merit but does not fully meet PLOS ONE’s publication criteria as it currently stands. Therefore, we invite you to submit a revised version of the manuscript that addresses the points raised during the review process.

We look forward to receiving your revised manuscript.

Kind regards,

Safdar Husain Tahir, PhD, Postdoc

Academic Editor

PLOS ONE

Reviewers' comments:

Reviewer's Responses to Questions

**Comments to the Author**

1. Is the manuscript technically sound, and do the data support the conclusions?

Reviewer #1: Yes

Reviewer #2: Partly

Reviewer #3: Partly

2. Has the statistical analysis been performed appropriately and rigorously? 

Reviewer #1: Yes

Reviewer #2: No

Reviewer #3: Yes

3. Have the authors made all data underlying the findings in their manuscript fully available?

Reviewer #1: Yes

Reviewer #2: No

Reviewer #3: No

4. Is the manuscript presented in an intelligible fashion and written in standard English?

Reviewer #1: Yes

Reviewer #2: No

Reviewer #3: Yes

5. Review Comments to the Author

Reviewer #1: I have evaluated article titled Composite Governance Mechanisms and Sustainable Economic Performance of Pakistan Textile Industry. I observe that this topic is important and interesting. While the topic is interesting, the paper requires revision to meet academic standards and enhance its appeal to a broader readership. Addressing these concerns will likely improve the manuscript’s chances of publication and its impact within the scholarly community, stock markets and other stakeholders.

Key concerns

Add justification for choosing Pakistan as a sample to clear the context.

The contribution section should be updated to reflect the paper's importance and uniqueness compared to existing literature. Including at least three new citations that relate to the work will help position the paper within the current research landscape.

The theoretical discussion in this paper is not enough and needs proper revision and development. While it reports on past studies in the introduction, it does not develop its own theoretical argument to support its main question. Moreover, I observe that this paper does not incorporate the theoretical implications of established corporate sustainable finance/standard finance. While discussing the topic of "Composite Governance Mechanisms and Sustainable Economic Performance of Pakistan Textile Industry," several theoretical frameworks could provide valuable support and context for the study. Kindly revisit and appropriately explain Upper Echelons Theory (UET), Managerial Network Theory (MNT), Resource-Based Theory (RBT) Stakeholder Theory, Institutional Theory, and Principal-Agent Theory. Additionally, discuss the theoretical implications of these theories in the results and discussion sections.

-Does corporate social responsibility mediate the relationship between corporate governance and firm performance? Empirical evidence from BRICS countries JCI 1.89IF(5) 10.6AJG 1SSCI Q1IF 9.8. W Akhter, A Hassan: Corporate Social Responsibility and Environmental Management 31 (1), 566-578

-Impact of boardroom diversity on corporate financial performance: T Bagh, MA Khan, NM Hammad Humanities and Social Sciences Communications (HSSCOM) 10 (222), 13. https://doi.org/10.1057/s41599-023-01700-3.

Update literature on some recent studies demonstrating role of corporate governance, ESG performance, and sustainable growth on various dimensions of corporate domains.

https://doi.org/10.1016/j.heliyon.2024.e26757,

https://doi.org/10.24136/oc.2023.036; https://doi.org/10.1016/j.bir.2024.04.005

In empirical model section, Author should cite the following publications and explain how GGM method is suitable in handling endogeneity issue. https://doi.org/10.1016/j.bir.2024.04.005.

I suggest that in the results section, create a heading “Baseline Estimation,” and under this heading, you should provide the interpretation of fixed effects. Authors should report either fixed effects or random effects based on the Hausman probability value. If the probability value is less than 0.05, only report fixed effect outcomes.

Then add heading Endogontiy Concerns and preset the GMM results

To ensure the robustness of the findings, additional robustness checks using alternative models (e.g., Feasible Generalized Least Squares) should be added. I suggest following stata code: xtgls dependent variablesindependent variables and controls variables

Or

xtscc dependent variable independent variables and controls variables i.year, fe

This paper lacks economic connectivity of findings and lacks quality discussion and implications of research, stating who will benefit and how and where it can be used further.

Proof- the paper properly and avoid using AI tools.

Reviewer #2: This manuscript considers the textile industry in Pakistan and attempts to provide an understanding of the relationship between corporate governance mechanisms and sustainable economic performance. Thank you for an interesting read.

Unfortunately, I don't believe this manuscript meets the criteria required for a research article in PLOS One. For instance, there were several cases in which the command of the English language was lacking. For example, on page 15 of 39: "The major responsibility of the AC is designing, overseeing, and implementing financial reporting practices of the company and, hence, ensuring better governance procedures". I fail to see how good financial reporting ensures better governance PROCEDURES. It may help ensure better overall governance, but how are governance PROCEDURES changed or affected by financial reporting? As another example, the in-text citation method is clunky and reduces readability considerably due to poor punctuation. Furthermore, the document is not self-consistent, as "Transparency and Disclosure" was abbreviated T&D and then later abbreviated using D&T. There are spacing issues (e.g., page 22 has "Akbar (2014)highlighted"). The use of "i.e.," was not done properly - it's "i-e" throughout the article. The services of a professional editor may help. Additionally, statistics can never "prove" anything - however, this phrasing was used many times throughout the text.

More worryingly, the connection between the various theories put forward from the literature and the current work was not clearly identified and fleshed out. Connections between paragraphs are lacking. A more substantive literature review, lending context to the current work instead of simply "They did __, They did ___" would be appropriate (e.g., page 19 of 39).

Of the 39 pages submitted, only 4 or so were related to methodology, and this was not well discussed. For example, "The study also eliminated sample components that were non-existent during the entire study period" - what does this mean? Which companies were removed from the original sample, and why? On page 21 of 39, "DER is the difference between the economic value generated and the economic value retained." Earlier, "economic value retained" was DER. So how can DER be the difference between DEG and DER? Further, "The study applied an item-wise disclosure measurement" - what is this and how is it calculated? The CGMI formula on page 22 is a bit odd to me. This creates a different percentage for each company, which may not be really fair. If company A scores 100%, and company B scores 100%, but company A had 25 factors and company B had 10, this formula considers company A and company B the same. They are clearly not. It may be best to weight this in a different way.

Section 3.3 has the empirical model, and this is where to explain how firm age and firm size were calculated. For example, the authors said firm age is the "number of years company experience since inception" (what does this mean? The number of years the company has been operating?) and firm size is "the logarithm of total assets". Why not just put "log(total assets)" in the model directly? And is this actually log_10, or is this ln? The abbreviation GMM is used without definition (I know it's generalized method of moments, but it can also stand for Gaussian mixture model, so please clarify). The last sentence indicates that unit root tests and the Durbin-Watson autocorrelation test were conducted, but the reason for these tests and what they accomplish was not provided. At the top of page 19, what is a "d-statistic"? Also, a null hypothesis cannot be "accepted" - it can only be rejected or not rejected.

Table 3 indicates a significance level (alpha) of 2.223. This is not possible, as the significance level must be a probability (so a value between 0 and 1). Table 4 lists p-values, but no confidence intervals. Please provide appropriate measures of variation.

Based on this and other vague descriptions (e.g., page 26: "The Hausman test determines the validity of models with p-values below 0.05, accepts H_1 and values greater than 0.05 accept H_0" - this is not a description of the test itself, as many statistical tests use 0.05 as a threshold), it's not clear that the authors understand their use of the statistical methods presented in this work.

I could go on, but suffice to say I believe this article requires quite a lot of revision. Perhaps with sufficient editing this may be brought to a venue other than PLOS One.

Reviewer #3: I appreciate the effort displayed by the authors to develop the manuscript. Hopefully, the comments provided will enhance the quality of the manuscripts.

In the abstract section, I will recommend that authors specifically state the model accepted by Hausman and complement it with the dynamic model, not simply state that fixed and random models were applied. Similarly, the abstract should be written in a clear and informative manner. 

The authors have made considerable efforts to develop the manuscript. Similarly, I have seen that the authors have highlighted the importance of the textile industry to the Pakistani economy, but they have not provided insights into the need for investigating the issue addressed. In particular, the motivation and problems that necessitate the investigation of carrying out the study are crucial.

Please check the citation properly.

In 4.3.3, while reporting the results, I think the reporting of your results should be based on which of the fixed and random effect models the Hausman test selected. This is then followed by a discussion of the GMM results.

Can the authors show the VIF results as well?

Please indicate the number of observations in the results tables and the constant values.

Table 3 can be merged with Table 4 or 5.

The composite measure can also be tested separately as an additional analysis in order to identify the impact of each component on sustainable economic performance.

The discussion and conclusion sections are not detailed enough. I suggest the authors should develop this section and align it with the theory that supports the findings.

The authors should also ensure that the writting is done in a professional manner.

6. PLOS authors have the option to publish the peer review history of their article (what does this mean?). If published, this will include your full peer review and any attached files.

Reviewer #1: No

Reviewer #2: No

Reviewer #3: **Yes: **Badru Bazeet Olayemi

---

## [Author Response · Author response to Decision Letter 0]

6 Sep 2024

Response Letter

The Editor, 

PLOS ONE JOURNAL

Manuscript ID # PONE-D-24-17105

Dear Dr. Safdar Husain Tahir,

Thank you for giving us an opportunity to revise our manuscript (PONE-D-24-17105), entitled “The Composite Governance Mechanisms and Sustainable Economic Performance of Pakistan’s Textile Industry” for publication consideration in your journal. We very much appreciate the constructive criticism provided by the reviewers. Now we have made a sustained effort to address each issue raised by all three reviewers and to improved our manuscript accordingly.

Here are the important amendments we have made to the manuscript:

All reviewers comments and our responses

Reviewer 1: Email Comments and our Responses

1. Thank you for the constructive comments. Now we have addressed all the mentioned concerns and improved the manuscript in line with suggestions of the distinguished referees. 

2. Justification for choosing Pakistan, specifically textile sector as a sample is added, please see paragraph 2 on page 4 in red color.

3. The study contributions are revised in line with the objectives of the study, an updated to reflect the paper's importance and uniqueness compared to existing literature. The new relevant citations are added, such as Afes and Anis, 2023); Achim et al. (2023); (Bui and Krajcsak, 2024), please see on page 6.

4. The theoretical discussion in the paper is revised and based upon the reporting of past studies, theoretical argument developed to support the main research question, please see section 2.1 theoretical lens on pages 7, 8, and 9.

Additionally, literature review section is revised and theoretical relevance is added on CGMI-performance relationship, please see pages 12, 13 and 14.

5. The theoretical contributions of the study are reviewed, five theoretical logics are incorporated: agency theory, stewardship theory, stakeholder theory, resource-based theory and transaction cost theory, adopted from (Bui and Krajcsak, 2024). Detail of theoretical lens follow a conceptual description of each theory, supported existing literature, and implications of the theory for study objectives. UET theory added in description, reference cited as (Augier & Teece, 2018); (Phan & Duong, 2021), please see page 9.

Secondly, the discussion and conclusion sections are revised, alignment of the findings with the past studies added. Theoretical implications are modified and restated to support the study findings, references cited as (Galbreath, 2018); (Munir et al. 2019); Madaleno and Vieira (2020); Benson & Ganda (2022); (Farooq et al., 2022); (Bui and Krajcsak, 2024), please see paragraph 2 on page 26 in red color

6. Thank you for your valuable suggestions. We have now reviewed all the recommended articles and incorporated relevant literature from recent studies. The references cited include (Iftikhar et al., 2024), (Kwarteng et al., 2023), and (Irshad et al., 2023). Please refer to page 14 for more details.

7. The mentioned publication added and reference cited as (Iftikhar et al., 2024), Please see page 25.

8. The results and discussion section are revised and a sequence followed by first presentation of Fixed effect estimation, Random effect estimation, Hausman test. Then endogeneity estimations satisfy the use of fixed and random effect model. Finally, GMM estimation model is applied for Robustness check. Instead of baseline estimation heading, a separate heading is added for each panel estimation model, please see page 23-25.

9. Limited past studies reference is available for implementation of FGLS in context of sustainable economic performance. We mentioned the implementation of other generalized least squares (GLS) Models: GMM, ARDL, and FGLS models, as recommendations for future research, please see page 29.

10. The whole conclusion and discussion section revised, followed by reporting of the study findings, alignment with the past studies and implications of the theories on the study findings are added, see on page 26,27, and 28.

Economic connectivity of the study findings is restated in context of minimizing agency cost and transaction cost.

Secondly, no AI content is incorporated writing the manuscript. Additionally, statement of Declaration of generative AI and AI-assisted technologies is added, see page 29.

Reviewer 2: Email Comments and our Responses

1. Thank you for the constructive comments. We have addressed all the mentioned concerns and improved the manuscript scope for possible publication in PLOS ONE.

A native English editor has polished the language to enhance clarity and readability.

2. With due respect, the major concern of the study is sustainable performance through better governance procedures, so the financial reporting is a least focused area. 

Perhaps this comment is the result of misinterpretation of our arguments. 

The findings concluded the significant impact of financial reporting disclosure and transparency on the corporate sustainable performance, please see page 27.

3. All the intext citations are corrected by applying APA 7th Edition referencing style.

The whole document is revised and punctuations are corrected by applying Grammarly software.

4. Now the whole document is revised and reviewed, specifically “Transparency and Disclosure".

Spacing issues are corrected, Use of i-e reviewed and amended by applying Grammarly Software.

Additionally, the sentences are reviewed and the phrases are replaced by a professional Editor. Now we are confident that the manuscript is up to the mark for PLOS ONE standard.

5. The theories mentioned in “the theoretical lens” section on pages 7, 8, and 9 are revised.

In addition, we have aligned the current work in the “literature review” section on pages 12, 13, 14 and 15 as well as coherence of concepts and the paragraphs are re-arranged. 

Now we have also discussed the strengths and weaknesses of different authors as well as critical analysis the current literature. 

Theoretical relevance from the existing studies is redescribed in the interpretation of results and discussion, kindly see section 4, pages 19 to 26.

6. Thanks for the valuable comment. Now the methodology section is revised, modified, and incorporated the necessary changes in each sub-heading. 

Kindly note, methodology section focused on four modules: sample selection and data sources, measurement variables, modeling equation for the proposed models, and description of estimation techniques.

The sample selection section is revised and criteria is restated and cited as (Elston, 2021; Saleem & Hashmi 2022), please see on pages 16, 17 and 18.

Please note, 26 companies were eliminated, and the panel consisted of total 98 firms. This is mentioned on page 16.

We have enhanced the methodological section as well as corrected confusing sentences by a professional native editor. 

7. Thank you for your careful review and for pointing out the errors in our manuscript. We have made the necessary corrections as follows:

Correction to DER Definition: We appreciate your observation regarding the inconsistency in the definition of DER. We have corrected this in the manuscript to clarify that DER represents the difference between DEG (Economic Value Generated) and DED (Economic Value Distributed).

Item-Wise Disclosure Measurement: We have revised the description of the item-wise disclosure measurement. As noted, it considers the mean value of all disclosure items, as detailed in Sobhani et al. (2012). Additionally, we have re-examined the references and updated the citations in the manuscript to reflect the most accurate sources: Ahmad et al. (2023) and Saleem & Hashmi (2022). Please refer to page 17 of the revised manuscript for these updates.

8. The mentioned CGMI formula is reviewed and corrected, referenced by (Munisi & Randoy, 2013) on page 17.

Secondly, A ‘Yes’ response shows the presence of element and is coded with 1 and a ‘No’ shows the absence of element and is coded with 0. The aggregate score for each sub-index is obtained averaging the items present divided by total no. of items and then CGMI score was calculated. 

The CGMI score obtained shows the implementation of corporate governance principles by each firm.

9. Section 3.2 “Descriptions of Variables” on page 17 is revised and relevant proxies are added for both firm size and firm age with cited references e.g., (Arora & Bodhanwala 2018; (Do et al., 2021); (Affes & Jarboui, 2023)

Kindly note, the study applied two control variables: firm size and firm age. Many current studies measure firm size by logarithm (ln) of total assets, and firm age by years of operating experience, e.g. (Arora & Bodhanwala 2018; Do et al., 2021; Affes & Jarboui, 2023). Please see on page 18 for more details. 

10. Now, the GMM abbreviation is revised and meaning added in both abstract and empirical relationships in section 2.3. Please see on page 13.

11. Thank you for pointing out the econometric errors in our manuscript. We have made the necessary corrections as follows:

Section 4.2.1 and 4.2.2 Serial Correlation and Panel unit root test are re-examined and interpretation of results are revised, please see on page 21 and 22.

12. Kindly note, the "d-statistic" is the Durbin-Watson statistic value, which measures the level of autocorrelation in the residuals.

Hence, the null hypothesis of no autocorrelation is not rejected. Please see more details on page 21.

Kindly note, Table 3 is corrected with modified description of “d statistic” Kindly see it on page 21. 

13. Thanks for the comment. Please note, the Table 4 shows the results of LLC, IPS and ADF test with P value. All the three tests for stationarity are reviewed, as there will be no confidence intervals.

Table 5, 6, 8 shows estimations for Fixed effect, Random effect and GMM estimations. In addition, we have re-calculated empirical estimation, confidence intervals in all three tables. The interpretations of the tables are also restated, please see on page 23, 24, 25 and 26.

Regarding the appropriate measures of variation, we have added r-square, Wald chi2 for each model. Please see tables 5, 6, 8 for more details.

14. The description of Fixed effect, Random effect and Hausman test are corrected, under section 3.4 “Panel Econometrics” on page 19 and 4.3.1. “Fixed Effect and Random Effect” on page 23 and GMM Model on page 25.

15. Thank you for your detailed feedback. We have carefully addressed all the suggestions you provided and have made substantial revisions to the manuscript. Additionally, a professional native speaker has polished the language to enhance clarity and readability We believe that these improvements have significantly strengthened the paper, making it appropriate for possible publication in PLOS ONE.

Reviewer 3: Email Comments and our Responses

1. Thank you for recognizing our efforts in developing the manuscript. We appreciate your thoughtful comments and have worked diligently to incorporate them, with the aim of further enhancing the quality of the paper.

2. Thank you for your kind suggestion. In response, we have revised the abstract to explicitly state the model accepted by the Hausman test and have included details about the dynamic model. We have also ensured that the abstract is now written in a clearer and more informative manner. Please see the abstract on page 1. 

3. Thank you for acknowledging our efforts in developing the manuscript. In response to your feedback, we have now provided a clearer justification for focusing on Pakistan's textile sector. We have highlighted the motivation and the specific challenges that make this investigation crucial. Please refer to “Introduction section” paragraph 2 on page 4 for these additions.

4. Thank you for pointing that out. We have thoroughly reviewed and corrected all in-text citations, ensuring they adhere to the APA 7th Edition referencing style.

5. In Section 4.3.1, we have revised the interpretation of results to align with the model selected by the Hausman test, followed by a discussion of the GMM results. We have also updated the reporting of results in Tables 6, 7, and 8, and revised the interpretation accordingly. Please refer to pages 22 to 25, where these changes are highlighted in red.

6. We have performed the VIF test, and the values are all below 5, indicating no issues with multicollinearity. The VIF results are provided on page 22.

7. The number of observations and constant values are corrected and added in the tables, please see table 2 (descriptive statistics) and table 3 serial correlation in section 4 “Results and Discussion” on page 27.

8. Table 4 is the results of Panel Unit root test, and table 7 shows Endogeneity test. Table 3 is the Durbin Watson serial correlation test, and experts recommend separate presentation provides better impact.

9. The GRI variable has been reviewed, and we have referenced a study by Ahmad et al. (2023), which supports the use of averaging the scores of each performance dimension to obtain a more accurate overall performance score. Please refer to page 17 for this discussion.

10. The entire conclusion and discussion sections have been revised. We have included a comprehensive report of the study findings, aligned them with previous research, and discussed the implications of the theories on our results. Please refer to pages 26, 27, and 28 for these updates.

11. A professional native speaker has polished the language to enhance clarity and readability.

12.

---

## [Editor Report · Decision Letter 1]

27 Sep 2024

PONE-D-24-17105R1The Composite Governance Mechanisms and Sustainable Economic Performance of Pakistan's Textile IndustryPLOS ONE

Dear Dr. Saleem,

Thank you for submitting your manuscript to PLOS ONE. After careful consideration, we feel that it has merit but does not fully meet PLOS ONE’s publication criteria as it currently stands. Therefore, we invite you to submit a revised version of the manuscript that addresses the points raised during the review process.

We look forward to receiving your revised manuscript.

Kind regards,

Safdar Husain Tahir, PhD, Postdoc

Academic Editor

PLOS ONE

Journal Requirements:

Additional Editor Comments:

Please provide a response to the reviewers, addressing each comment with corresponding page and paragraph numbers. Ensure that all changes made in the main document are highlighted clearly.

---

## [Author Response · Author response to Decision Letter 1]

6 Oct 2024

The Editor, 

PLOS ONE JOURNAL

Manuscript ID # PONE-D-24-17105R1

Asima Saleem

National University of Modern Languages, Islamabad

Pakistan

Email: asimas009@mail.com

Dated: 3-10-2024

Dear Dr. Safdar Husain Tahir,

Thank you for giving us an opportunity to revise our manuscript (PONE-D-24-17105R1), entitled “The Composite Governance Mechanisms and Sustainable Economic Performance of Pakistan’s Textile Industry” for publication consideration in your journal. We very much appreciate the constructive criticism provided by the reviewers. Now we have made a sustained effort to address each issue raised by all three reviewers and to improved our manuscript accordingly.

Here are the important amendments we have made to the manuscript:

Journal Requirements

Response: The following suggestions are incorporated and addressed as:

The reference list has been thoroughly reviewed to ensure that all citations are complete, correct, and up-to-date.

We conducted a search to identify if any of the cited articles have been retracted. None of the references in the manuscript have been found to be retracted. 

For manuscript improvement, some relevant, current references are added to maintain the scientific integrity of the manuscript, cited as (Tahir et al., 2014), (Tahir et al., 2016), (Saleem et al., 2021), (Raza et al., 2023), (Tahir et al., 2024), (Tahir et al., 2019), (Chandani & Ahmed, 2021). Please see (page 3 para 2), (page 6 para 1), (page 8 para 2), (page 10 para 2), (page 14 para 2), highlighted in red.

All the modifications to the reference list, are added in the rebuttal letter, with a detailed explanation of pages and paragraphs for the changes.

Additional Editor Comments:

Please provide a response to the reviewers, addressing each comment with corresponding page and paragraph numbers. Ensure that all changes made in the main document are highlighted clearly.

Response: The editor comments are incorporated and modified in the current reviewer’s response sheet and rebuttal letter as well.

The response to each comment is modified with corresponding page and paragraph numbers. All the changes made in the document are highlighted in red.

Reviewer 1: Email Comments and our Responses

i. I have evaluated article titled Composite Governance Mechanisms and Sustainable Economic Performance of Pakistan Textile Industry. I observe that this topic is important and interesting. While the topic is interesting, the paper requires revision to meet academic standards and enhance its appeal to a broader readership. Addressing these concerns will likely improve the manuscript’s chances of publication and its impact within the scholarly community, stock markets and other stakeholders.

Response: Thank you for the constructive comments. Now we have addressed all the mentioned concerns and improved the manuscript in line with suggestions of the distinguished referees. 

ii. Add justification for choosing Pakistan as a sample to clear the context.

Response: Justification for choosing Pakistan, specifically textile sector as a sample is added, please see paragraph 2 on page 4 in red color.

iii. The contribution section should be updated to reflect the paper's importance and uniqueness compared to existing literature. Including at least three new citations that relate to the work will help position the paper within the current research landscape.

Response: The study contributions are revised in line with the objectives of the study, an updated to reflect the paper's importance and uniqueness compared to existing literature. The new relevant citations are added, such as Afes and Anis, 2023); Achim et al. (2023); (Bui and Krajcsak,2024), please see on page 6 paragraph 3.

iv. The theoretical discussion in this paper is not enough and needs proper revision and development. While it reports on past studies in the introduction, it does not develop its own theoretical argument to support its main question.

Response: The theoretical discussion in the paper is revised and based upon the reporting of past studies, theoretical argument developed to support the main research question, please see section 2.1 theoretical lens on pages 7 (Paragraph 3), page 8 (Paragraph 1 and 2), and page 9 (Paragraph 1 and 2).

Additionally, literature review section is revised and theoretical relevance is added on CGMI-performance relationship, please see page 12 (paragraph 2), page 13 (paragraph 2) and page 14 (paragraph 2), page 15 (paragraph 1).

v. Moreover, I observe that this paper does not incorporate the theoretical implications of established corporate sustainable finance/standard finance. While discussing the topic of "Composite Governance Mechanisms and Sustainable Economic Performance of Pakistan Textile Industry," several theoretical frameworks could provide valuable support and context for the study. Kindly revisit and appropriately explain Upper Echelons Theory (UET), Managerial Network Theory (MNT), Resource-Based Theory (RBT), Stakeholder Theory, Institutional Theory, Principal-Agent Theory

Additionally, discuss the theoretical implications of these theories in the results and discussion sections.

Response: The theoretical contributions of the study are reviewed, five theoretical logics are incorporated: agency theory, stewardship theory, stakeholder theory, resource-based theory and transaction cost theory, adopted from (Bui and Krajcsak, 2024). Detail of theoretical lens follow a conceptual description of each theory, supported existing literature, and implications of the theory for study objectives. UET theory added in description, reference cited as (Augier & Teece, 2018); (Phan & Duong, 2021), please see page 9 (paragraph 2).

Secondly, the discussion and conclusion sections are revised, alignment of the findings with the past studies added. Theoretical implications are modified and restated to support the study findings, references cited as (Galbreath, 2018); (Munir et al. 2019); Madaleno and Vieira (2020); Benson & Ganda (2022); (Farooq et al., 2022); (Bui and Krajcsak, 2024), please see paragraph 1 on page 27 in red color

vi. Does corporate social responsibility mediate the relationship between corporate governance and firm performance? Empirical evidence from BRICS countries JCI 1.89IF(5) 10.6AJG 1SSCI Q1IF 9.8. W

Akhter, A Hassan: Corporate Social Responsibility and Environmental Management 31 (1), 566-578

-Impact of boardroom diversity on corporate financial performance: T Bagh, MA Khan, NM Hammad Humanities and Social Sciences Communications (HSSCOM) 10 (222), 13. https://doi.org/10.1057/s41599-023-01700-3.Update literature on some recent studies demonstrating role of corporate governance, ESG performance, and sustainable growth on various dimensions of corporate domains.

https://doi.org/10.1016/j.heliyon.2024.e26757

https://doi.org/10.24136/oc.2023.036

https://doi.org/10.1016/j.bir.2024.04.005

Response: Thank you for your valuable suggestions. We have now reviewed all the recommended articles and incorporated relevant literature from recent studies. The references cited include (Iftikhar et al., 2024), (Kwarteng et al., 2023), and (Irshad et al., 2023). Please refer to page 15 (paragraph 1) for more details.

vii. In empirical model section, Author should cite the following publications and explain how GGM method is suitable in handling endogeneity issue. 

https://doi.org/10.1016/j.bir.2024.04.005.

Response: The mentioned publication added and reference cited as (Iftikhar et al., 2024), Please see page 26 Paragraph 1.

viii. I suggest that in the results section, create a heading “Baseline Estimation,” and under this heading, you should provide the interpretation of fixed effects. Authors should report either fixed effects or random effects based on the Hausman probability value. If the probability value is less than 0.05, only report fixed effect outcomes.

Then add heading Endogeneity Concerns and preset the GMM results

Response: The results and discussion section are revised and a sequence followed by first presentation of Fixed effect estimation, Random effect estimation, Hausman test. Then endogeneity estimations satisfy the use of fixed and random effect model. Finally, GMM estimation model is applied for Robustness check. Instead of baseline estimation heading, a separate heading is added for each panel estimation model, please see page 23-25 (paragraph 1 and 2).

ix. To ensure the robustness of the findings, additional robustness checks using alternative models (e.g., Feasible Generalized Least Squares) should be added. I suggest following stata code: xtgls dependent variables independent variables and controls variables Or xtscc dependent variable independent variables and controls variables i.year, fe

Response: Limited past studies reference is available for implementation of FGLS in context of sustainable economic performance. We mentioned the implementation of other generalized least squares (GLS) Models: GMM, ARDL, and FGLS models, as recommendations for future research, please see page 29 paragraph 2.

x. This paper lacks economic connectivity of findings and lacks quality discussion and implications of research, stating who will benefit and how and where it can be used further.

Proof- the paper properly and avoid using AI tools.

Response: The whole conclusion and discussion section revised, followed by reporting of the study findings, alignment with the past studies and implications of the theories on the study findings are added, see on page 27, and 28 (paragraph 2).

Economic connectivity of the study findings is restated in context of minimizing agency cost and transaction cost.

Secondly, no AI content is incorporated writing the manuscript. Additionally, statement of Declaration of generative AI and AI-assisted technologies is added, see page 30.

Reviewer 2: Email Comments and our Responses

xi. This manuscript considers the textile industry in Pakistan and attempts to provide an understanding of the relationship between corporate governance mechanisms and sustainable economic performance. Thank you for an interesting read.

Response: Thank you for the constructive comments. We have addressed all the mentioned concerns and improved the manuscript scope for possible publication in PLOS ONE.

xii. Unfortunately, I don't believe this manuscript meets the criteria required for a research article in PLOS One. For instance, there were several cases in which the command of the English language was lacking. For example, on page 15 of 39: "The major responsibility of the AC is designing, overseeing, and implementing financial reporting practices of the company and, hence, ensuring better governance procedures".

Response: Grammarly software original version is applied for all grammatical, punctuation, readability, consistency, and sentence structuring errors. 

Additionally, an English literature scholar from National University of Modern Languages has edited the language, proofreading and necessary English language editing.

xiii. I fail to see how good financial reporting ensures better governance PROCEDURES. It may help ensure better overall governance, but how are governance PROCEDURES changed or affected by financial reporting?

Response: With due respect, the major concern of the study is sustainable performance through better governance procedures, so the financial reporting is a least focused area. 

Perhaps this comment is the result of misinterpretation of our arguments. 

The findings concluded the significant impact of financial reporting disclosure and transparency on the corporate sustainable performance, please see page 28 (paragraph 1).

xiv. As another example, the in-text citation method is clunky and reduces readability considerably due to poor punctuation.

Response: All the intext citations are corrected by applying APA 7th Edition referencing style.

The whole document is revised and punctuations are corrected by applying Grammarly software.

xv. Furthermore, the document is not self-consistent, as "Transparency and Disclosure" was abbreviated T&D and then later abbreviated using D&T. There are spacing issues (e.g., page 22 has "Akbar (2014)highlighted"). The use of "i.e.," was not done properly - it's "i-e" throughout the article. The services of a professional editor may help. Additionally, statistics can never "prove" anything - however, this phrasing was used many times throughout the text.

Response: Now the whole document is revised and reviewed, specifically “Transparency and Disclosure".

Spacing issues are corrected, Use of i-e reviewed and amended by applying Grammarly Software.

Additionally, the sentences are reviewed and the phrases are replaced by a professional Editor. Now we are confident that the manuscript is up to the mark for PLOS ONE standard.

xvi. More worryingly, the connection between the various theories put forward from the literature and the current work was not clearly identified and fleshed out. Connections between paragraphs are lacking. A more substantive literature review, lending context to the current work instead of simply "They did __, They did ___" would be appropriate (e.g., page 19 of 39).

Response: The theories mentioned in “the theoretical lens” section are revised, please see on page 7 (paragraph 3), page 8 (paragraph 1 and 2), and page 9 (paragraph 1 and 2) 

In addition, we have aligned the current work in the “literature review” section on pages 12, 13, 14 and 15 as well as coherence of concepts and the paragraphs are re-arranged. 

Now we have also discussed the strengths and weaknesses of different authors as well as critical analysis the current literature. 

Theoretical relevance from the existing studies is redescribed in the interpretation of results and discussion, kindly see section 4, pages 22 (paragraph 1), page 23 (paragraph 1 and 2), page 24 (paragraph 1), page 24 (paragraph 1), page 26 (paragraph 1 and 2), page 27 (paragraph 1).

xvii. Of the 39 pages submitted, only 4 or so were related to methodology, and this was not well discussed. For example, "The study also eliminated sample components that were non-existent during the entire study period" – what does this mean? Which companies were removed from the original sample, and why? 

Response: Thanks for the valuable comment. Now the methodology section is revised, modified, and incorporated the necessary changes in each sub-heading. 

Kindly note, methodology section focused on four modules: sample selection and data sources, measurement variables, modeling equation for the proposed models, and description of estimation techniques.

The sample selection section is revised and criteria is restated and cited as (Elston, 2021; Saleem & Hashmi 2022), please see on page 16 (paragraph 2 and 3), page 17 (paragraph 1) and page 18 (paragraph 1).

Please note, 26 companies were eliminated, and the panel consisted of total 98 firms. This is mentioned on page 16 (paragraph 3), page 17 (paragraph 1).

We have enhanced the methodological section as well as corrected confusing sentences by a professional native editor. 

xviii. On page 21 of 39, "DER is the difference between the economic value generated and the economic value retained." Earlier, "economic value retained" was DER. So how can DER be the difference between DEG and DER? Further, "The study applied an item-wise disclosure measurement"

Response: Thank you for your careful review and for pointing out the errors in our manuscript. We have made the necessary corrections as follows:

Correction to DER Definition: We appreciate your observation regarding the inconsistency in the definition of DER. We have corrected this in the manuscript to clarify that DER represents the 

---

## [Decision Letter · Decision Letter 2]

11 Dec 2024

The Composite Governance Mechanisms and Sustainable Economic Performance of Pakistan's Textile Industry

PONE-D-24-17105R2

Dear Dr. Asima Saleem,

We’re pleased to inform you that your manuscript has been judged scientifically suitable for publication and will be formally accepted for publication once it meets all outstanding technical requirements.

Kind regards,

Imran Anwar

Academic Editor

PLOS ONE

---

## [Editor Report · Acceptance letter]

15 Jan 2025

PONE-D-24-17105R2 

PLOS ONE

Dear Dr. Saleem, 

I'm pleased to inform you that your manuscript has been deemed suitable for publication in PLOS ONE. Congratulations! Your manuscript is now being handed over to our production team.

Kind regards, 

on behalf of

Dr. Imran Anwar 

Academic Editor

PLOS ONE